# An Explainable Radiogenomic Framework to Predict Mutational Status of KRAS and EGFR in Lung Adenocarcinoma Patients

**DOI:** 10.3390/bioengineering10070747

**Published:** 2023-06-21

**Authors:** Berardino Prencipe, Claudia Delprete, Emilio Garolla, Fabio Corallo, Matteo Gravina, Maria Iole Natalicchio, Domenico Buongiorno, Vitoantonio Bevilacqua, Nicola Altini, Antonio Brunetti

**Affiliations:** 1Department of Electrical and Information Engineering, Polytechnic University of Bari, Via Orabona 4, 70126 Bari, Italy; berardino.prencipe@poliba.it (B.P.); c.delprete4@studenti.poliba.it (C.D.); domenico.buongiorno@poliba.it (D.B.); nicola.altini@poliba.it (N.A.); antonio.brunetti@poliba.it (A.B.); 2Department of Medical and Surgical Sciences, University of Foggia, Viale Pinto 1, 71122 Foggia, Italy; emiliogarolla@gmail.com (E.G.); fabiocor@hotmail.it (F.C.); 3Molecular Oncology and Pharmacogenomics Laboratory, University of Foggia, Viale Pinto 1, 71122 Foggia, Italy; iole.nat@tiscali.it; 4Apulian Bioengineering SRL, Via delle Violette 14, 70026 Modugno, Italy

**Keywords:** lung adenocarcinoma, CT, KRAS, EGFR, radiogenomics, explainability

## Abstract

The complex pathobiology of lung cancer, and its spread worldwide, has prompted research studies that combine radiomic and genomic approaches. Indeed, the early identification of genetic alterations and driver mutations affecting the tumor is fundamental for correctly formulating the prognosis and therapeutic response. In this work, we propose a radiogenomic workflow to detect the presence of KRAS and EGFR mutations using radiomic features extracted from computed tomography images of patients affected by lung adenocarcinoma. To this aim, we investigated several feature selection algorithms to identify the most significant and uncorrelated sets of radiomic features and different classification models to reveal the mutational status. Then, we employed the SHAP (SHapley Additive exPlanations) technique to increase the understanding of the contribution given by specific radiomic features to the identification of the investigated mutations. Two cohorts of patients with lung adenocarcinoma were used for the study. The first one, obtained from the Cancer Imaging Archive (TCIA), consisted of 60 cases (25% EGFR, 23% KRAS); the second one, provided by the Azienda Ospedaliero-Universitaria ’Ospedali Riuniti’ of Foggia, was composed of 55 cases (16% EGFR, 28% KRAS). The best-performing models proposed in our study achieved an AUC of 0.69 and 0.82 on the validation set for predicting the mutational status of EGFR and KRAS, respectively. The Multi-layer Perceptron model emerged as the top-performing model for both oncogenes, in some cases outperforming the state of the art. This study showed that radiomic features can be associated with EGFR and KRAS mutational status in patients with lung adenocarcinoma.

## 1. Introduction

Lung cancer is the leading cause of cancer-related deaths worldwide. Its complex pathobiology has prompted research studies that combine radiomic and genomic approaches. According to the International Agency for Research on Cancer (IARC), there were an estimated 2.21 million cases of lung cancer in 2020, making it the second most common cancer after breast cancer (2.26 million cases) and ahead of colorectal cancer (1.93 million cases). However, lung cancer remains the most common cause of cancer death globally, with an estimated 1.80 million deaths in 2020 [1]. Lung adenocarcinoma is a clinically relevant disease due to its high incidence and prevalence. It is the most common form of non-small cell lung cancer (NSCLC), accounting for approximately 40% of all NSCLC diagnoses, which in turn represents about 85% of all lung cancer cases [2]. Typically, lung adenocarcinoma develops in the periphery of the lung and may be discovered in areas of chronic inflammation or scars [3]. Adenocarcinoma is known to have a higher incidence among non-smokers than other forms of NSCLC, and it tends to spread to lymph nodes and other organs, making it a serious disease with significant clinical implications in diagnosis, treatment, and patient follow-up. Its high morbidity and mortality are primarily attributed to the often-late diagnosis at an advanced stage when curative treatment options are limited [4]. The 5-year survival rate for patients with metastatic lung carcinoma is generally low at 22% [5].

Recent studies have shown that lung adenocarcinoma has a complex genomic landscape, with multiple genetic alterations and driver mutations that may affect tumor initiation, progression, and response to therapy [6]. KRAS and EGFR mutations are common in this form of cancer and can influence treatment response. Specifically, KRAS mutations are associated with resistance to conventional chemotherapy drugs used for lung adenocarcinoma [7]. EGFR mutations, instead, are present in a significant percentage of non-small cell lung adenocarcinoma patients and are associated with a higher response to EGFR receptor inhibitors such as gefitinib, erlotinib, and afatinib [8]. Notably, the presence of these oncogenic mutations is mutually exclusive; patients display differing clinical and pathological features, as well as prognostic and predictive implications, depending on the specific mutation.

Due to its spread, the early diagnosis of this cancer and its characterization have a high impact on human health. In fact, the discovery of specific treatments for patients with EGFR mutation has significantly impacted the clinical treatment of lung cancer, since these patients have demonstrated nearly twice the duration of survival compared to those without it [9].

Medical imaging techniques can offer a non-invasive way to visualize phenotypic differences between different kinds of tumors and lesions. In this context, radiomics is a rapidly expanding approach that involves extracting a large number of quantitative features from images to determine the phenotype of regions of interest, possibly through the use of intelligent automatic systems [10,11,12,13,14]. These features encompass a range of properties, including shape characteristics, textural features, and pixel intensities and can provide information about areas affected by a disease, also allowing the identification and quantitative description of tumor patterns and characteristics [15,16,17,18].

Radiomic features provide information about both the grey-level patterns and inter-pixel relationships in medical images; spectral and shape properties can be extracted from the regions of interest as well. These features can then be used to create computational models based on machine learning techniques, which can aid in the diagnosis, prognosis, and treatment guidance of various medical conditions [19,20,21]. Certain radiomic features have even been found to identify genetic alterations in tumor DNA, which has resulted in the development of the field of radiogenomics [22], which strives to find correlations between image features and characteristics of the genome of the patient. Such a process aims to unveil the dynamics involved at the genetic and molecular levels through medical image analysis. Indeed, computational techniques to extract quantitative features from those images can be linked to distinctive characteristics of the phenotype and genotype of the tissue. Radiomics has garnered significant attention in the context of lung cancer screenings with the aim of enhancing diagnostic sensitivity and specificity while reducing the time burden placed on radiologists. Additionally, radiomics is being utilized in precision medicine to forecast prognosis and the efficacy of specific therapies [23].

In this work, we first propose a radiogenomic workflow to characterize lung adenocarcinoma cases with respect to KRAS and EGFR mutational status using radiomic features extracted from the computed tomography (CT) images of patients affected by lung adenocarcinoma. Then, we aimed to enhance the understanding of radiomic features associated with KRAS and EGFR mutations in lung adenocarcinoma and to evaluate the effectiveness of the features in accurately classifying mutations. To accomplish this, we employed the SHAP (SHapley Additive exPlanations) technique for explainability, which allowed us to gain insights into the contribution of individual features with respect to the classification outcomes. Figure 1 shows the pipeline of the implemented workflow.

In a previous study, we developed and validated a pipeline based on the radiomic approach to quantitatively characterize and classify lung nodules as either adenocarcinoma or other histological classes, using unenhanced CT images [10]. In this work, we considered adenocarcinoma only, and we focused on the classification of KRAS and EGFR mutations.

With respect to existing literature, the innovative contributions of this manuscript can be summarized as follows:A realization of a radiogenomic pipeline to characterize CT images of patients with EGFR and KRAS mutations;A systematic comparison of feature selection techniques to enhance the accuracy and reliability of the developed models;An explainability analysis to corroborate machine learning findings with existing studies in radiology.

The remainder of this work is organized as follows: Section 2 describes the state of the art of the detection of EGFR and KRAS mutations by means of radiomic workflows; Section 3 details the datasets, the radiomic features, the feature-selection techniques, the machine learning models adopted for classifying the occurrence of mutations, and the explainability framework employed to reveal the feature significance. The results are presented in Section 4 and discussed in Section 5. Final remarks and directions for future works are reported in Section 6.

## 2. Related Works

Ongoing studies on the molecular profile of cancer have revealed that EGFR and KRAS play a role in the development, invasiveness, and metastasis of non-small cell lung cancer. Consequently, these mutations are given primary consideration in supporting treatment decisions for NSCLC [24].

Mei et al. [25] focused on identifying a correlation between radiomic features derived from CT images of lung adenocarcinoma and EGFR mutation. Logistic regression analysis identified younger age and the feature named ‘Grey Level Non-Uniformity Normalized’ as predictors for exon 19 mutation; age, female gender, and the feature named ’Maximum 2D Diameter Column’ for exon 21 mutation, and female gender, non-smoking status, and the feature named ’Size Zone Non-uniformity Normalized’ for EGFR mutations. The area under the receiver operator characteristic curve (AUC) of the combination of clinical and radiomic features for predicting EGFR mutations was 0.66.

Shiri et al. [26] conducted a study that investigated the effectiveness of a radiomic framework in predicting the mutational status of EGFR and KRAS in patients with NSCLC. The study utilized radiomic features extracted from low-dose CT, contrast-enhanced diagnostic quality CT (CTD), and Positron Emission Tomography (PET) imaging modalities. A total of 150 NSCLC patients were involved in the research, and six feature-selection methods and twelve classifiers were employed for the multivariate prediction of the gene mutational status. To evaluate the developed models, an independent validation set consisting of 68 patients was used. The results demonstrated that the inclusion of radiomic features significantly enhanced the accuracy of predicting EGFR and KRAS mutational status compared to conventional PET parameters.

In a study by Le et al. [27], a machine-learning-based model was proposed for the selection of radiomic features and prediction of EGFR and KRAS mutations in NSCLC patients. The study involved a cohort of 161 patients with NSCLC from the Cancer Imaging Archive (TCIA), out of 211 patients initially selected, and analyzed low-dose CT images for detecting EGFR and KRAS mutations. Different feature selection analyses were performed to identify radiomic features that might be crucial for the proposed machine-learning-based model, such as univariate selection, recursive feature elimination, feature importance, filter methods, F-score, genetic algorithm, minimum redundancy feature selection, and the KBest algorithm. The performances of the proposed models were evaluated using a validation set consisting of 18 patients derived from the same TCIA dataset. The results showed that the genetic algorithm plus the XGBoost classifier exhibited the most favorable performance, with an AUC of 0.89 and 0.81 for detecting EGFR and KRAS mutations, respectively.

Jia et al. [24] considered two cohorts of patients, the first consisting of 345 patients, of whom 60.8% were positive for the EGFR mutation, and the validation cohort consisting of 158 patients, 57.6% with EGFR mutation. The AUC obtained by the Random Forest trained using 94 features was 0.80, which increased to 0.83 with the addition of clinical information, including sex and smoking history.

Rios et al. [28] aimed to develop a radiomic model for predicting EGFR mutations in 258 patients with lung adenocarcinoma from CT images. The authors identified the most informative radiomic and semantic features using minimum-redundancy–maximum-relevance (MRMR) feature selection. They developed and evaluated logistic regression models using a training set of 172 patients and an independent validation set of 86 patients, with the AUC used as a metric for performance evaluation. After the MRMR feature selection, the combined radiomic–semantic logistic regression model exhibited the best performance, with a validation AUC of 0.67. Meanwhile, the radiomic and semantic models had AUCs of 0.56 and 0.63, respectively.

The study conducted by Liu et al. [29] aimed to assess the ability of radiomic features extracted from CT scans to predict EGFR mutation status in patients with surgically resected peripheral lung adenocarcinomas in an Asian cohort. A total of 298 patients were included in this retrospective study, and 219 quantitative 3D features were extracted from segmented tumor volumes, with 59 independent features included in the analysis. Mutant EGFR was found to be significantly associated with the female sex, never-smoker status, lepidic predominant adenocarcinomas, and low or intermediate pathological grade. Univariate analysis showed statistically significant differences in 11 radiomic features between the EGFR mutant and wild-type groups. A set of 5 radiomic features (i.e., CT attenuation energy, tumor main direction, and texture defined by wavelets and laws) were able to predict mutant EGFR status with an AUC of 0.65. Furthermore, the addition of radiomic features to a clinical model resulted in a significant improvement in predictive power, with the AUC increasing from 0.67 to 0.71.

Pinheiro et al. [30] used the NSCLC-Radiogenomics dataset, which comprises CT images and molecular data collected from 211 patients with NSCLC, to predict the EGFR and KRAS mutation status using radiomic and semantic features extracted from the CT images, along with clinical features. The results showed that the predictive models for EGFR had the best performance, with a maximum mean AUC of 0.75 obtained using hybrid semantic features.

Moreno et al. [31] proposed an ensemble approach using a voting scheme, i.e., Selective Class Average Voting, to improve EGFR and KRAS mutation prediction. The machine learning approach allowed increasing the AUC from 0.68 to 0.70 for EGFR mutation and 0.65 to 0.71 for KRAS mutation.

## 3. Materials and Methods

In this section, we describe the materials and the methods employed throughout this study. First, the datasets are presented in Section 3.1. Then, radiomic features are outlined in Section 3.2. The techniques employed for feature selection are listed in Section 3.3, and the classification workflow is portrayed in Section 3.4. Finally, the explainability methodology is reported in Section 3.5.

### 3.1. Datasets

In this work, we referred to two different cohorts of NSCLC patients. For the first cohort, namely D1, CT, and PET images, the gene-expression microarray, RNA-seq, and clinical data of 211 different patients were available [32]. However, to specifically target cases of lung adenocarcinoma, we considered a subset of 60 individuals characterized by segmented images of lesions and the mutation panel.

The second dataset, namely D2, has been collected from Azienda Ospedaliero–Universitaria “Ospedali Riuniti” of Foggia and contained CT images, the lesion segmentation masks, and the mutation panels of 55 patients.

### 3.2. Radiomic Features

Pathological processes can influence the organ structure and also cause an alteration in their shape and texture. A way to assess these characteristics is radiomics, which can be defined as the computation of quantitative descriptors which could allow both the local and global characterization of regions of interest (ROIs).

The radiomic features are usually categorized into different classes, each analyzing specific characteristics of the region. There are first-order features, which describe the distribution of voxel intensity in the area. Morphological features, instead, characterize the geometrical aspect of the region. The textural-based features describe the spatial correlation of the intensity levels, exploiting several mathematical structures, such as the Gray Level Co-occurrence Matrix (GLCM) [33], the Gray Level Size Zone Matrix (GLSZM) [34,35], the Gray Level Run Length Matrix (GLRLM) [36], the Neighboring Gray Tone Difference Matrix (NGTDM) [37], and the Gray Level Dependence Matrix (GLDM) [38]. The features can also be extracted from the images after the application of several filters, such as the Laplacian of Gaussian and Wavelet transform.

Some of the most critical challenges in radiomics are the lack of reproducibility and validation. To overcome these issues, an international organization, the Image Biomarker Standardization Initiative (IBSI) [17], endowed a standard definition of the radiomic features. In this work, we adopted PyRadiomics [39], a Python library in compliance with the IBSI, to extract an initial feature set of 1772 radiomic features for both D1 and D2 datasets.

### 3.3. Feature Selection

After the extraction of the initial set of 1772 features, we conducted a univariate analysis on the D1 dataset in order to assess the significance of the features in distinguishing between the wildtype or mutant status of KRAS and EGFR genes. Several Feature Sets (FS) were considered and used to construct the datasets, which were input to the machine learning classifiers. A summary of the FS is available in Table 1. Specifically,

**FS1** was obtained as follows: models for univariate binary logistic regression were trained for the two genes on the input features after being normalized in terms of z-score. The features with a *p*-value >0.05 were discarded, and only the uncorrelated features were retained, by exploiting the algorithm described in Bevilacqua et al. [15]. The correlation between quantitative features was estimated with the Pearson correlation coefficient.**FS2** was obtained in the same way as **FS1** but considering only features with a *p*-value <0.01 and without exploiting the algorithm [15] to retain the uncorrelated features.**FS3** and **FS4** were obtained in the same way as **FS1** and **FS2**, respectively, but without performing the preliminary z-score normalization.**FS5** was obtained in the same way as **FS3** but applying Benjamini–Hochberg correction for the *p*-values.**FS6** was obtained by applying the Mann–Whitney U test to calculate the *p*-values. Subsequently, the features with a *p*-value >0.05 were discarded.**FS7** was obtained by applying a two-sided t-test after feature normalization to calculate the *p*-values. Subsequently, the features with a *p*-value >0.05 were discarded.

To perform the subsequent prediction task, each FS was computed referring to the EGFR and KRAS genes; by doing this, we obtained the feature sets **FSiE**, for the first case, and **FSiK**, for the latter, where i=1,⋯,7 denotes the index of the specific FS.

### 3.4. Prediction of KRAS and EGFR Mutational Status

The aim of this study was to classify the mutational status of KRAS and EGFR comparing several machine learning classifiers trained on radiomic features extracted from CT images. To do this, we employed Logistic Regression (LR), Support Vector Machines (SVM), Adaptive Boosting (AB), Random Forest (RF), Multi-layer Perceptron (MLP), XGBoost (XGB), Gradient Boost Model (GBM), Gaussian Naive Bayes (GNB), and an ensemble (ENS) of all these models, which was implemented using a soft voting classifier. These models were selected due to their wide use in radiomic studies for medical classification purposes [15,40,41,42]. The models’ performances were evaluated on the AUC computed on the validation set. Figure 2 depicts the workflow employed in this study for training the classification models.

To increase the performance of the classifiers trained on the FS, we also evaluated seven different Feature Set Collections (FSCs) for training and evaluating the machine learning models that predict the mutational status of EGFR and KRAS:**FSC1**: FS1E, FS1K, FS2E, FS2K, FS3E, FS3K, FS4E, and FS4K were merged to predict both EGFR and KRAS;**FSC2**: FS1E, FS2E, FS3E, and FS4E were merged to predict EGFR; FS1K, FS2K, FS3K, and FS4K were merged to predict KRAS;**FSC3**: FS2E and FS4E were merged to predict EGFR; FS2K and FS4K were merged to predict KRAS;**FSC4**: FS1E and FS2E were merged to predict EGFR, and FS1K BS FS2K were merged to predict KRAS;**FSC5**: FS3E and FS4E were merged to predict EGFR, and FS3K and FS4K were merged to predict KRAS;**FSC6**: FS6E and FS7E were merged to predict EGFR, and FS6K and FS7K were merged to predict KRAS;**FSC7**: FS1E, FS2E, FS3E, FS4E, FS5E, FS6E, and FS7E were merged to predict EGFR, and FS1K, FS2K, FS3K, FS4K, FS5K, FS6K, and FS7K were merged to predict KRAS.

In order to evaluate the performances of the classifiers and the reliability of the radiomic features, we conducted two experiments for each FSC:**Experiment 1**: We trained the model on the publicly available dataset (D1) and tested on the external dataset (D2) to evaluate its ability to generalize on new data;**Experiment 2**: We performed internal cross-validation by merging D1 and D2 to assess the performance difference with respect to external validation. We employed a 10-fold cross-validation with a stratified method to ensure an even distribution of classes across each fold.

For both experiments, we employed a grid search to determine the optimal hyperparameters for each model, exhaustively evaluating all the possible combinations of hyperparameters detailed in Table 2. For the final model evaluation, only the optimal configuration of hyperparameters has been considered.

### 3.5. Radiogenomic Features Explainability

To provide an explanation for the output generated by the machine learning models based on the radiomics feature sets, we conducted an analysis with the Shapley Additive Explanation (SHAP) tool [43]. SHAP is a model explanation framework for machine learning models that assists in evaluating the relative importance of different features in the model’s decision-making process. It is based on the concept of the ‘Shapley value’, borrowed from cooperative game theory [44]. SHAP aims to provide individualized explanations for model predictions, helping to identify the features that made the most significant contribution to a specific prediction and assessing how a particular feature influenced the overall prediction outcome.

Only the top-performing classification models for predicting the mutational status of both KRAS and EGFR were considered for this analysis since feature interpretability from best-performing models is more meaningful.

## 4. Results

In this section, we present the results of our study. First, the results of the univariate analysis are presented in Section 4.1. Then, the results of the classification models are reported in Section 4.2.

### 4.1. Univariate Analysis

In the first phase of the study, a univariate analysis was conducted to select the most significant features for the classification of KRAS and EGFR mutations. The statistical techniques considered for univariate analysis were binary logistic regression, Mann–Whitney U test and *t*-test, as reported in Table 1. A total of 1772 features were extracted from D1; then, different pipelines were used to select the most relevant ones. Specifically:**FS1E** resulted in 4 features with a *p*-value <0.05 and correlation coefficient <0.5;**FS1K** resulted in 3 features with a *p*-value <0.05 and a correlation coefficient <0.5;**FS2E** resulted in 1 feature with a *p*-value less than 0.01;**FS2K** did not result in the selection of any significant feature.**FS3E** resulted in 14 features with a *p*-value <0.05 and a correlation coefficient <0.5;**FS3K** resulted in 13 features with a *p*-value <0.05 and a correlation coefficient <0.5;**FS4E** resulted in 13 features with a *p*-value <0.01 and a correlation coefficient <0.5;**FS4K** resulted in 12 features with a *p*-value <0.01 and a correlation coefficient <0.5;**FS5E** resulted in 14 features with a *q*-value <0.05 and a correlation coefficient <0.5;**FS5K** resulted in 14 features with a *q*-value <0.05 and a correlation coefficient <0.5;**FS6E** resulted in 3 features with a *p*-value <0.05;**FS6K** resulted in 6 features with a *p*-value <0.05;**FS7E** resulted in 2 features with a *p*-value <0.05;**FS7K** resulted in 3 features with a *p*-value <0.05.

To facilitate the identification of clusters and assess the separability of the features belonging to wildtype and mutant classes for both genes, we applied Uniform Manifold Approximation and Projection (UMAP) [45] and t-distributed Stochastic Neighbor Embedding (t-SNE) [46] to the FSC2, since it showed the highest AUC value in the prediction of the mutational status. Figure 3 shows the embeddings for EGFR and KRAS mutations obtained by applying UMAP and t-SNE techniques.

### 4.2. Predictive Models

The findings from each experiment are presented in Figure 4 and Figure 5, where each heatmap shows the AUC values obtained on the validation set per model and input FS and FSC, respectively. Since univariate analyses selected sets of, in some cases overlapping, features, the dimensionality of the input to predictive models changed. In particular, the number of features for predicting EGFR was 15 for FSC1; 18 for FSC2; 14 for FSC3; 5 for FSC4; 13 for FSC5; 9 for FSC6; 28 for FSC7. For KRAS: 23 for FSC1; 17 for FSC2; 12 for FSC3; 3 for FSC4; 14 for FSC5; 5 for FSC6; and 20 for FSC7.

The aim of Experiment 1 was to evaluate the discriminative power of the radiomic signature obtained from D1, following the implemented feature reduction approach, for classifying the mutational status of KRAS and EGFR. To this purpose, the classification models were trained on the sets of features obtained through the univariate analyses performed on D1. The results obtained in this experiment on the prediction of the mutational status of EGFR and KRAS are shown in Figure 4a,c, respectively, for what concerns FS. In the case of FSC, results are displayed in Figure 5a,c for EGFR and KRAS, respectively.

The MLP model with a hidden layer of 20 neurons achieved the highest AUC value of 0.69 in the prediction of the mutational status of EGFR. For KRAS, the top-performing model was MLP, with a hidden layer of 10 neurons with an AUC value of 0.82. In both cases, MLP obtained the best result by exploiting the FSC2. The SHAP analysis was conducted on the top model with the highest AUC values in predicting the mutational status of both EGFR and KRAS.

Experiment 2, instead, aimed to evaluate the performance of the models when D1 and D2 were merged. However, AUCs obtained via internal cross-validation were inferior compared to those of Experiment 1. The results obtained for predicting the mutational status of EGFR and KRAS in this experiment are presented in Figure 4b,d, respectively, for what concerns FS. In the case of FSC, results are displayed in Figure 5b,d for EGFR and KRAS, respectively. The ENS model achieved an AUC value of 0.71 for predicting the mutational status of EGFR, exploiting the FSC1. Among single models, the GNB achieved an AUC of 0.70 with the FSC7. On the other hand, for what concerns KRAS, different combinations of feature sets and feature set collections for RF, XGB, GBM managed to achieve an AUC of 0.68.

Generally, we can note that there is huge variability in the results of Experiment 1 (external validation setting). This is because the two datasets present inherent differences, and not all machine learning models are adequately capable of generalizing on unseen data. On the other hand, the results of Experiment 2 (cross-validation setting) are closer among different models. Indeed, in this case, examples of both datasets are also in the training set, and it is easier to transfer knowledge to the validation set.

Figure 6 shows the SHAP analysis related to the best models (MLPs, AUC = 0.69 and AUC = 0.82) for the prediction of the mutational status of EGFR and KRAS, respectively. For EGFR, the features that contributed to obtaining that results were ‘Original firstorder 10Percentile’ and the other two derived from images filtered with Laplacian of Gaussian and wavelet transform, namely ‘High Gray Level Run Emphasis’ obtained from GLRLM and ‘Large Area Low Gray Level Emphasis’ obtained from GLSZM, respectively. The SHAP values of ‘Original firstorder 10Percentile’ showed a mainly positive effect on the model prediction for values close to zero, but a negative effect for higher values. This suggests a non-linear relationship between this feature and EGFR mutation. This could be due to the higher density of tumor tissue compared to surrounding normal tissue, leading to lower pixel intensities in normal tissue and a low value of ’Original firstorder 10Percentile’. A high value of ‘log sigma 2.5 mm 3D glrlm HighGrayLevelRunEmphasis’ indicates the presence of homogeneous and coherent regions of high gray-level intensity within the ROI. For predicting the EGFR mutation, other important features were ‘wavelet HLL glszm Large Area Low Gray Level Emphasis’ and ‘wavelet HHL glszm Large Area Low Gray Level Emphasis’, which describe the emphasis of large low gray-level areas in the image, calculated using the GLSZM matrix on the HLL and HHL sub-bands of the wavelet-decomposed image, respectively. These features indicate the presence of large homogeneous and coherent regions of low gray-level intensity in the image. The SHAP values of these features provide insights into their impact on model predictions. When the SHAP values are concentrated around 0 and low for positive values, it indicates that these features do not strongly influence model predictions when their quantity increases. However, they may still have an impact on model predictions when present in smaller quantities or in combination with other features.

The ’original shape SurfaceArea’ feature was crucial for predicting the KRAS mutation in the model, which led to an AUC of 0.82. High values of this feature had a positive impact and contributed significantly to the prediction of the KRAS mutation, as shown in Figure 6.

## 5. Discussion

Our study involved a comprehensive comparison of feature selection and machine learning approaches to efficiently detect EGFR and KRAS mutations in patients with lung adenocarcinoma using radiomic features extracted from CT images. The aim of this study was to enhance the comprehension of radiomic features linked with KRAS and EGFR mutations in lung adenocarcinoma and evaluate the effectiveness of these features in accurately classifying mutations. The accurate identification of patients with KRAS or EGFR mutations is critical for the planning of individualized therapeutic strategies, as these mutations have been shown to play a significant role in cancer progression and response to treatment.

Two cohorts of patients with lung adenocarcinoma were used for the study. The first one, obtained from TCIA, consisted of 60 cases (25% EGFR, 23% KRAS); the second one, provided by the Azienda Ospedaliero-Universitaria ’Ospedali Riuniti’ of Foggia, was composed of 55 cases (16% EGFR, 28% KRAS).

The first phase of the study involved the extraction of radiomic features from the original CT images, the images filtered with Laplacian of Gaussian and Wavelet transformation. The 1772 radiomic features extracted underwent a univariate analysis to evaluate their importance and significance. Several classification models and feature selection techniques were implemented to predict the mutational status of KRAS and EGFR. We observed a wide range of performances for predictive models using various radiomics features, feature selection techniques, and classification algorithms. For EGFR mutation status prediction, the best predictive power was achieved by MLP, with an AUC of 0.69 on the validation set. This model was trained using significant features obtained from FSC2. For KRAS, the best predictive power was achieved by MLP, with an AUC of 0.82 on the validation set. Additionally, in this case, the second Feature Set Collection led to the best result.

Pinheiro et al. [30], Shiri et al. [26], and Le et al. [27] analyzed radiomic features extracted from the same public dataset included in the current work and developed different approaches for feature selection and classification of KRAS and EGFR mutational status. Pinheiro et al. developed a model based on the XGBoost algorithm and its feature-ranking function. They selected 37 significant radiomics features, resulting in an AUC of 0.75. However, their model failed to detect KRAS mutation, with an AUC of 0.50. Shiri et al. developed a Stochastic Gradient Descent classifier, achieving an AUC of 0.83 for detecting KRAS mutation and an AUC of 0.78 for detecting EGFR mutation, using features extracted from low-dose CT, PET, and contrast-enhanced diagnostic quality TC. Le et al. used low-dose CT images’ radiomic features to detect EGFR and KRAS mutations. The results showed that genetic algorithms for feature selection and the XGBoost classifier exhibited the best performance with an AUC of 0.89 for EGFR and 0.81 for KRAS.

The studies mentioned above employed radiomic features extracted from various medical images such as CT, CDT, and PET in patients with NSCLC. However, none of them utilized an external dataset to validate the proposed classification models. Hence, the generalization capability of the developed classifiers was not thoroughly assessed. In contrast, the present study considered an external dataset for validation.

Mei et al. [25], Jia et al. [24], Rios et al. [28], and Liu et al. [29] employed machine learning models for predicting the mutational status of EGFR in patients affected by lung adenocarcinoma, using a different publicly available dataset than ours. The best results were achieved by Jia et al., with an AUC of 0.82, using a Random Forest model trained with 94 radiomic features and clinical data. The studies conducted by Mei et al., Rios et al., and Liu et al. yielded AUC values ranging from 0.66 to 0.71. In contrast to our study, none of these studies, which focus on patients with lung adenocarcinoma have incorporated feature selection models and machine learning algorithms for predicting the mutational status of KRAS. Furthermore, these studies employed different types of data, i.e., CT images, clinical and semantic data.

The models proposed in our study achieved an AUC of 0.69 and 0.82 on the validation set for predicting the mutational status of EGFR and KRAS, respectively. Notably, the MLP model trained with FSC2 emerged as the top-performing model in both cases. A comprehensive comparison between the proposed datasets and models’ results with respect to the ones considered in the literature analysis is reported in Table 3.

Finally, the use of SHAP revealed that ‘original firstorder 10Percentile’, ‘log sigma 2.5 mm 3D glrlm HighGrayLevelRunEmphasis’, and ‘wavelet HLL glszm Large Area Low Gray Level Emphasis’ are among the most important features for predicting EGFR mutation in the top-performing model, while ‘original shape SurfaceArea’, ‘original firstorder 10Percentile’, and ‘original firstorder Maximum’ are important for the prediction of KRAS mutational status.

## 6. Conclusions

In this paper, we presented an explainable radiogenomic workflow to characterize EGFR and KRAS mutations from the CT scans of lung adenocarcinoma patients.

Overall, this study demonstrated the potential of radiomic features and machine learning in predicting EGFR and KRAS mutations in lung adenocarcinoma patients. The findings contribute to advancing personalized treatment strategies for these patients by enabling precise identification of mutational status.

The main advantage of the proposed pipeline is that it allows realizing a quantitative and repeatable assessment of radiomic characteristics, which represents an improvement over manual inspection of CT scans from expert radiologists. On the other hand, more data and further studies are required to implement such decision-support systems in the clinical routine practice.

Future research should further explore the utility of SHAP and incorporate larger and more diverse datasets to enhance the robustness and generalizability of predictive models for these mutations.

## Figures and Tables

**Figure 1 bioengineering-10-00747-f001:**
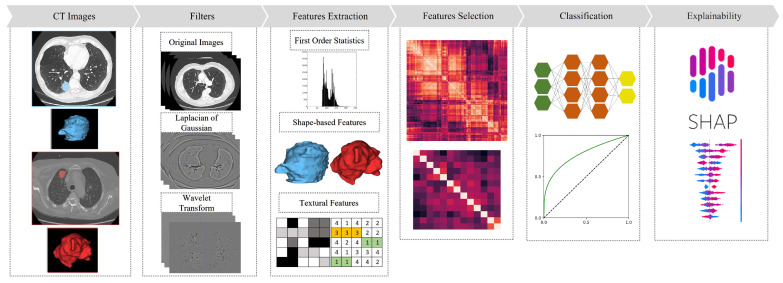
Proposed Radiogenomics Workflow. The first stage of our workflow involves the acquisition of unenhanced CT images and corresponding mutational data. Features are then extracted from images and filtered versions of the images. Machine learning classifiers are trained on top of these features. Lastly, SHAP is employed to understand the rationale behind classifiers’ decisions.

**Figure 2 bioengineering-10-00747-f002:**
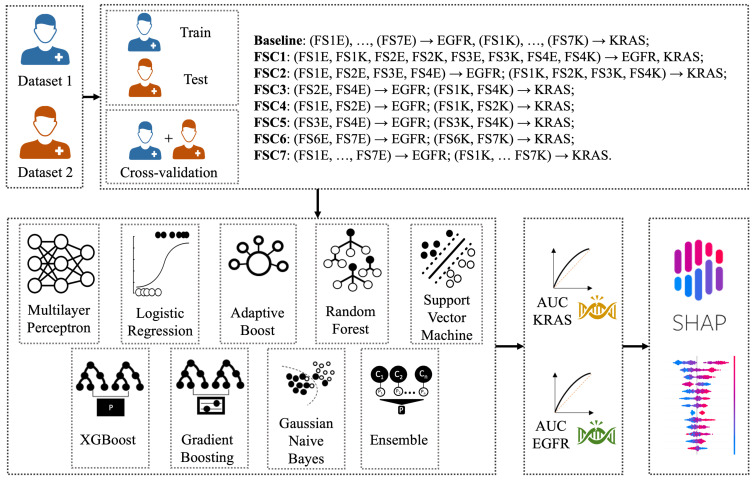
The workflow of the classification process was employed for predicting EGFR and KRAS mutations. The baseline involved using only the FS for predicting the EGFR and KRAS mutation status. The FSC was obtained by merging different FSs.

**Figure 3 bioengineering-10-00747-f003:**
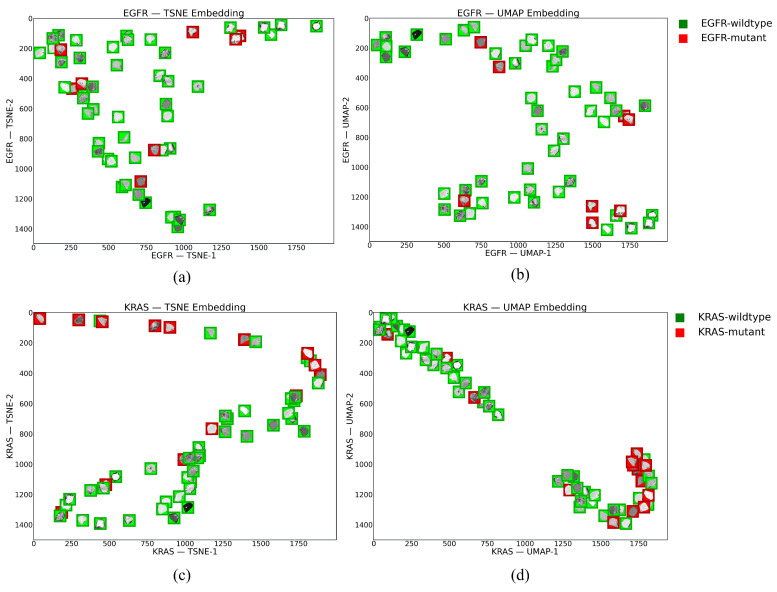
t-SNE and UMAP were obtained considering the FSC2, which resulted in the best AUC values for predicting the mutational status of EGFR and KRAS. (**a**,**b**) show t-SNE and UMAP visualizations for EGFR, respectively. (**c**,**d**) show t-SNE and UMAP visualizations for KRAS, respectively.

**Figure 4 bioengineering-10-00747-f004:**
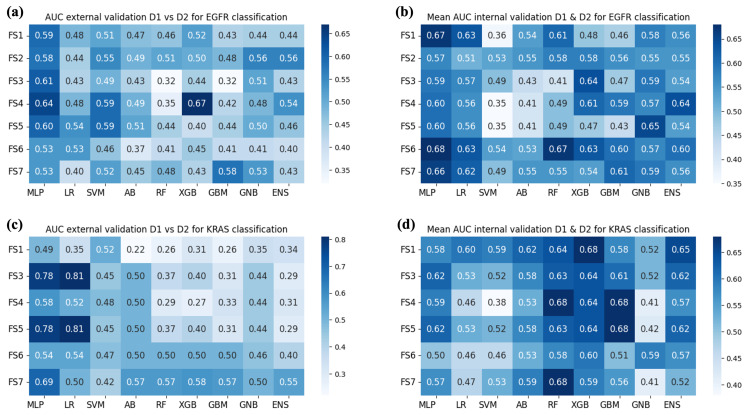
AUC obtained by the classification models used for each considered FS. (**a**) Results of Experiment 1—EGFR mutation. (**b**) Results of Experiment 2—EGFR mutation. (**c**) Results of Experiment 1—KRAS mutation. (**d**) Results of Experiment 2—KRAS mutation.

**Figure 5 bioengineering-10-00747-f005:**
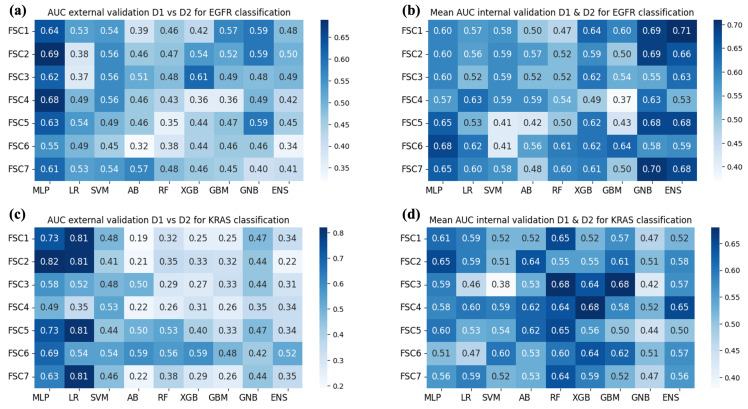
AUC obtained by the classification models used for each considered FSC. (**a**) Results of Experiment 1—EGFR mutation. (**b**) Results of Experiment 2—EGFR mutation. (**c**) Results of Experiment 1—KRAS mutation. (**d**) Results of Experiment 2—KRAS mutation.

**Figure 6 bioengineering-10-00747-f006:**
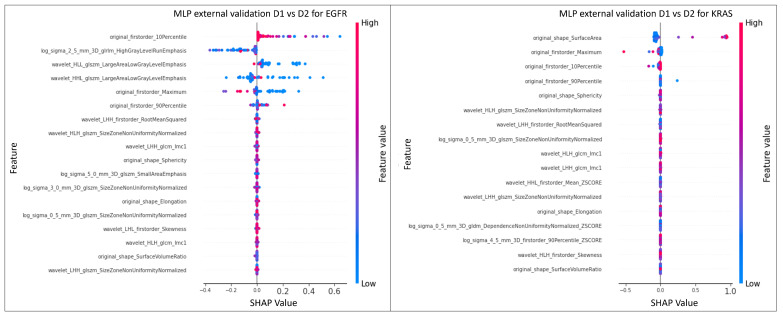
SHAP of the models that achieved the best AUC value on the validation dataset for EGFR and KRAS classification.

**Table 1 bioengineering-10-00747-t001:** Feature sets were constructed from the initial radiomic features. Each row represents a specific method used to obtain a particular feature set. In particular, we compared different combinations of normalization techniques, statistical significance methods, *p*-value correction approaches, *p*-value thresholds, and correlation coefficient thresholds to remove correlated features.

Feature Set	Normalization	*p*-Value			*p*-Value Correction	*p*-Value Threshold		Correlation Coefficient
		Logistic Regression	Mann-Whitney U Test	*t*-Test		0.05	0.01	
FS1	**✓**	**✓**	**✗**	**✗**	**✗**	**✓**	**✗**	**✓**
FS2	**✓**	**✓**	**✗**	**✗**	**✗**	**✗**	**✓**	**✗**
FS3	**✗**	**✓**	**✗**	**✗**	**✗**	**✓**	**✗**	**✓**
FS4	**✗**	**✓**	**✗**	**✗**	**✗**	**✗**	**✓**	**✓**
FS5	**✗**	**✓**	**✗**	**✗**	**✓**	**✓**	**✗**	**✓**
FS6	**✗**	**✗**	**✓**	**✗**	**✗**	**✓**	**✗**	**✗**
FS7	**✓**	**✗**	**✗**	**✓**	**✗**	**✓**	**✗**	**✗**

**Table 2 bioengineering-10-00747-t002:** Training hyperparameters for each considered machine learning classifier. Names of hyperparameters are in conformity with the scikit-learn library. Parameters inside brackets have undergone a grid search.

Model	Hyperparameters	Value
LR	max_iter	2500
	C	1
	penality	[‘l1’, ‘l2’]
	solver	[‘liblinear’, ‘saga’]
SVM	kernel	‘rbf’
	C	[0.1, 1, 10]
	gamma	[0.01, 0.1, 1]
RF	n_estimators	[10, 50, 100]
	min_samples_split	[2, 5, 10]
	min_samples_leaf	[1, 2, 4]
	criterion	‘Gini’
	max_depth	[5, 10, 15, 20]
AB	n_estimators	[50, 100, 150]
	learning_rate	[0.01, 0.1, 1.0]
MLP	hidden_layer_sizes	[(5), (10), (20), (30), (5, 5), (10, 5), (10,10), (20,5), (20,10), (30,10), (20, 20), (30,20)]
	activation	‘Relu’
	solver	‘Adam’
	alpha	0.0001
	learning_rate_init	0.001
	max_iter	2500
	early_stopping	False
XGB	n_estimators	[50, 100, 150]
	learning_rate	[0.01, 0.1, 1]
GBM	n_estimators	[50, 100, 150]
	learning_rate	[0.01, 0.1, 1]
GNB	priors	None
	var_smoothing	$10^{-9}$

**Table 3 bioengineering-10-00747-t003:** Description of studies aimed at predicting the mutational status of KRAS and EGFR in patients with NSCLC using different types of data, with T representing the training set and V representing the validation set in the EGFR and KRAS mutations columns. LUAD stands for lung adenocarcinoma.

Author	Data	Study Population	EGFR Mutation Frequency	KRAS Mutation Frequency	AUC EGFR	AUC KRAS	External Cohort
Shiri et al. [26]	CT, CTD, PET	NSCLC	25% (T) 25% (V)	25% (T) 23% (V)	0.82	0.83	**✓**
Jia et al. [24]	CT, Clinical	LUAD	61% (T) 58% (V)	**✗**	0.82	**✗**	**✓**
Rios et al. [28]	CT, Semantic	LUAD	45%	**✗**	0.67	**✗**	**✗**
Liu et al. [29]	CT, Clinical	LUAD	46%	**✗**	0.71	**✗**	**✗**
Le et al. [27]	CT, Clinical	NSCLC	18%	19%	0.89	0.81	**✗**
Pinheiro et al. [30]	CT, Clinical, Semantic	NSCLC	20%	23%	0.58	0.50	**✗**
Moreno et al. [31]	CT, Clinical	NSCLC	14%	24%	0.70	0.71	**✗**
Mei et al. [25]	CT, Clinical	LUAD	51%	**✗**	0.66	**✗**	**✗**
Ours	CT	LUAD	23% (T) 13% (V)	25% (T) 24% (V)	0.69	0.82	**✓**

## Data Availability

The data presented in this study are available upon request from the corresponding author.

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
