# Peer review of "An Explainable Radiogenomic Framework to Predict Mutational Status of KRAS and EGFR in Lung Adenocarcinoma Patients"

_bioengineering, 2023, doi:10.3390/bioengineering10070747_

Round 1

Reviewer 1 Report

The manuscript is interesting and very well written. However, here are some suggestions that can slightly improve the quality of the manuscript. 

1. At the end of the introduction section, clearly indicate the hypotheses of your research and scientific contribution. These two should be formatted in bullet form. At the end of the introduction section write the paragraph describing the outline of the paper. This paragraph improves the readability of the manuscript. 

2. Write a paragraph between the section and subsection. For example between Materials and Methods and subsection dataset. This paragraph should contain basic information about the section in this case materials and methods section .

3.  In Table 2 the hyperparameters should have the same font format as the manuscript 

4. Figure 3 should contain plot grids. By doing so you are improving the readability of the plots 

5. Discussion and Conclusion should be separated. The discussion section should contain a discussion about obtained results and a comparison to the previous research.  The Conclusion section should contain four paragraphs. 

First paragraph: Basic description of what was done in the paper, 

Second paragraph: The answers to the hypotheses defined in the introduction section based on provided discussion section 

Third paragraph: Description of advantages and disadvantages of the proposed method. 

Fourth paragraph: Possible directions for the future work. 

6. Why did you employ the 10-fold cross-validation? Why not 5 or 3 fold cross-validaiton? 

7. It would be better if you have used the enseble i.e. combined all algorithms in the ensemble and used Decision tree as final estimator. 

Additional comments: First time you use the abbreviation in the manuscript is should contain the full name followed with the abbreviation in brackets. For example in abstract MLP: 

Multi-Layer Perceptron (MLP) 

Every other time you should use the MLP. 

English language is fine. 

Reviewer 2 Report

Up to date topic

The article is well developed and well written

Figures and Tables are explanatory and of great quality

Reviewer 3 Report

From a biostats/clinical epidemiology point of view, this manuscript has been well planned and realized. Of note, several topics have been "teached" to the readers, thus enhancing their knowledge of radiomics and ML applied to medical oncology! Some minor comments:

- line 11 TCIA, undefined

- line 211, better to say univariate binary logistic regression, since multinomial logistic models do exist too

- line 213, I do suggest to use a non-parametric approach for the estimation of correlation coefficients, actually this info has not been reported

- line 232, what about to add some other classical ML approaches, like GBM, NB, XGBoost and DNN!? Have you estimated any stacked ML model too, could it be of help in this context!?

- line 281, univariate analysis should be more properly defined, since many different of them do exist!

- figures 4 and 5, I do believe that more complete comments on AUCs values and significance could improve this section

minor

Round 2

Reviewer 1 Report

The paper is improved according to the suggestions. It can be published in this form. 

No comment. English is fine.